# Temporally Generalizable Land Cover Classification: A Recurrent Convolutional Neural Network Unveils Major Coastal Change through Time

Patrick Clifton Gray [1],* , Diego F. Chamorro [2] , Justin T. Ridge [1] , Hannah Rae Kerner [3] , Emily A. Ury [4] and David W. Johnston [1]

1 Duke University Marine Laboratory, Nicholas School of the Environment, Duke University, Beaufort, NC 28516, USA; justin.ridge@duke.edu (J.T.R.); david.johnston@duke.edu (D.W.J.)
2 Department of Electrical and Computer Engineering, Duke University, Durham, NC 27708, USA; diego.chamorro@duke.edu
3 Department of Geographical Sciences, University of Maryland, College Park, MD 20742, USA; hkerner@umd.edu
4 Department of Biology, Duke University, Durham, NC 27708, USA; emily.ury@duke.edu
* Correspondence: patrick.c.gray@duke.edu

**Abstract:** The ability to accurately classify land cover in periods before appropriate training and validation data exist is a critical step towards understanding subtle long-term impacts of climate change. These trends cannot be properly understood and distinguished from individual disturbance events or decadal cycles using only a decade or less of data. Understanding these long-term changes in low lying coastal areas, home to a huge proportion of the global population, is of particular importance. Relatively simple deep learning models that extract representative spatiotemporal patterns can lead to major improvements in temporal generalizability. To provide insight into major changes in low lying coastal areas, our study (1) developed a recurrent convolutional neural network that incorporates spectral, spatial, and temporal contexts for predicting land cover class, (2) evaluated this model across time and space and compared this model to conventional Random Forest and Support Vector Machine methods as well as other deep learning approaches, and (3) applied this model to classify land cover across 20 years of Landsat 5 data in the low-lying coastal plain of North Carolina, USA. We observed striking changes related to sea level rise that support evidence on a smaller scale of agricultural land and forests transitioning into wetlands and "ghost forests". This work demonstrates that recurrent convolutional neural networks should be considered when a model is needed that can generalize across time and that they can help uncover important trends necessary for understanding and responding to climate change in vulnerable coastal regions.

**Keywords:** Landsat; land cover classification; coastal change detection; temporal generalization; deep learning; recurrent convolutional neural network; sea level rise; wetland monitoring

## 1. Introduction

Low lying coastal areas, home to a huge proportion of the global population, are locations where understanding long term changes in land cover is of particular societal importance. Ecologically, these regions are particularly affected by urban and agricultural expansion, stochastic natural events such as hurricanes, and chronic stressors such as sea level rise (SLR) [1]. The pressure of SLR in particular is submerging and adding salt to low lying land, one conspicuous result of this saltwater intrusion is the creation of "ghost forests". Although rates of conversion to ghost forests are not well understood, this is a widespread change of ecological and economic importance [2]. Forests are a crucial component of global carbon, nutrient, and hydrologic cycles [3]; they create vast wildlife habitat across coastal regions and provide important ecosystem services to communities across

the globe [4]. Their degradation can lead to long-term impacts on global biogeochemical cycles [5] as well as immediate impacts on local air quality, water quality, and climate [6,7].

The southeast United States (US) has a considerable low lying coastal plain that is home to major population centers, large swaths of farmland, and ecologically rich forests and wetlands [8]. Across the coastal plain, a large proportion of the land area is within one meter of sea level (Figure 1). The low elevation, high connectivity to water, and frequent substantial storm surge associated with hurricanes all enhance SLR impacts in this region. Together, these factors make saltwater intrusion a major threat to agriculture, forest ecosystems, and local communities [2,9,10]. In the Southeast US and in similar areas across the globe, forest die-off and dramatically altered biogeochemical cycles caused by SLR may be a dire positive feedback loop in our changing climate.

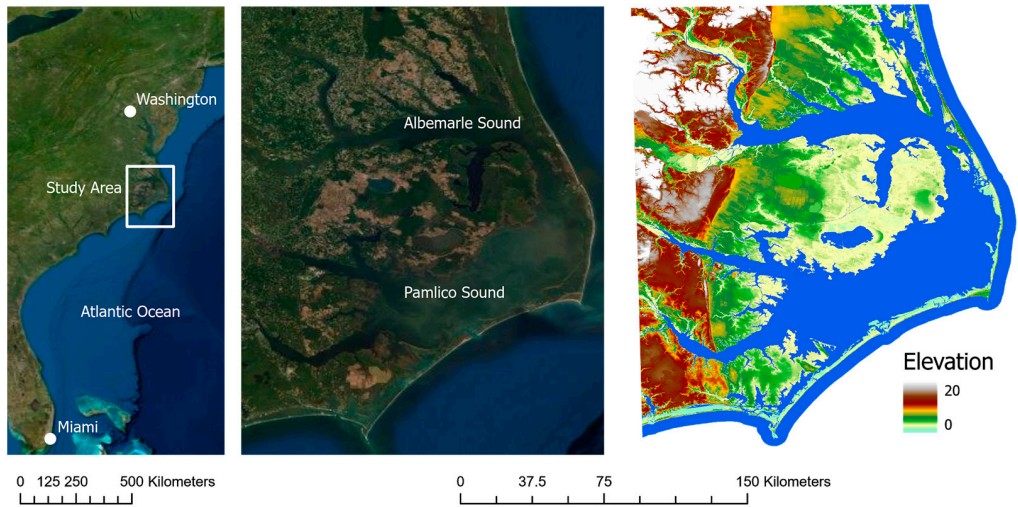

**Figure 1.** The North Carolina coastal plain is shown both in true color and a LiDAR digital elevation model (referenced to the North American Vertical Datum of 1988).

As we work towards a better understanding of these changes, accurately classifying land cover in decades past, before appropriate training and validation data exist, would improve our knowledge of subtle long-term impacts of climate change in these low-lying areas. This longer time series could help distinguish these trends from singular disturbance events (e.g., hurricanes) or decadal cycles (e.g., el Niño). While there is a substantial body of work on extending the length of remote sensing based land cover time series (e.g., [11]), many supervised discrete classification approaches that require accurate training data (i.e., ground-truth land cover reference data) are limited by the availability of this training data to the 2000s onward and cannot properly leverage the rich Earth observation record of Landsat spanning back into the early 1980s.

While land cover classification has always been a focus of remote sensing, high intra-class variance and low inter-class variance continue to challenge traditional approaches [12]. The noisiness of the data and the overlap of class spectral signatures is a major obstacle in the pursuit of models that can generalize to periods when reliable training data is not available. When temporal generalization is attempted, classification performance is typically much lower or even unreported for time periods not captured in the training data. This issue impedes important investigations of land cover dynamics. New deep learning models that properly extract representative spatiotemporal patterns could lead to major improvements in temporal generalizability, permitting classification back to the beginning of the satellite record.

## 1.1. Classifying Land Cover across Time

Detecting and predicting changes in land cover has been a key activity of remote sensing since the very origins of the field [12]. Despite considerable effort and methodolog-

ical improvements [13] the heterogeneity that is characteristic of the Earth's natural and built surface make this a challenging classification problem. Many approaches exist for monitoring change across time (see [14] for a Landsat-specific review of approaches). One common approach to land cover classification across years is to run a yearly classification across many years and filter out anomalous changes that likely result from misaligned phenological states and data artifacts [15]. LandTrendr is a widely used workflow that analyzes Landsat spectral trajectories on a per-pixel basis to identify transition points and long-term trends among noisy trajectories [16]. More recent work on disturbance monitoring extends what has primarily been a forest focused approach and is able to detect spectral breaks indicative of disturbance in different ecosystems across the Landsat archive [17]. The National Land Cover Database (NLCD) is a major effort to produce US-wide land cover maps every three years that is based on a decision tree classifier with substantial post-processing, in which the primary inputs are Landsat imagery and a digital elevation model [18]. While this post-processing improves the output, it is a huge effort that prevents frequent classification and rapid product availability [19].

Decision tree and random forest based approaches are common, but neural networks have been under consideration for over two decades, with early approaches finding moderate success by ingesting the same pixel location in two different images and predicting a binary output of change or no-change [20]. Spectral signal generalization (or extension), aiming to derive consistent spectral signatures regardless of image, environmental, and phenological conditions is helpful for using irregularly timed images but still faces challenges to fully incorporate the myriad of possible changes [21]. Change vector analysis examines the magnitude and direction of the change in spectra, which can be made discrete by determining a threshold that constitutes a change. This can be effective when images are properly radiometrically normalized and has been used successfully in combination with a clustering-based classification across Canada's natural parks [22]. The radiometric normalization that facilitates class signatures extension through space and time is a challenge in and of itself with previous work showing that normalization for successful signature extension should account for changes in the sensor itself, phenology, and sun-surface-sensor geometry [23]. Growth-stage normalization schemes have been developed that permit time series classification in the face of domain shifts, either from climate change or human policies [24]. Often simpler vegetation indices and phenological metrics are used to characterize land cover change rather than performing discrete classification [25]. Another approach is to develop a single reference map and to change this map to a future or past state with expert knowledge of feasible transitions and time series of spectra [26]. While there are a diverse set of continually improving approaches to accurately classifying land cover classification across space and time, substantial pre and post-processing, expert knowledge, and workflows focused on specific ecosystem changes can make models challenging to apply, overly sensitive to irrelevant changes in the data, and in many cases accuracy is often lower than desired to monitor subtle changes [27].

The long record of Landsat is increasingly suitable for identifying subtle land cover changes driven by climate change. However, poor classification accuracy of land cover maps that span long time periods remains a roadblock to capitalizing on this observation record. While this is not true of all classes or types of change, increases in accuracy and robustness to irrelevant noise are needed to improve understanding of subtle changes across long periods of time. If resolved, long time series of land cover could be critical in monitoring coastal regions and facilitating their management.

A considerable issue with classifying land cover across time stems from the fact that the spectral and spatial appearance of a single class changes throughout the year. This is exacerbated when monitoring across years because the phenological conditions, which are themselves not precisely equal from year to year, influence model prediction. While this can be confounding if only a single moment in time is considered by the model, this intra-annual phenological change can itself contain critical information for distinguishing classes if these patterns can be incorporated into the model [15,28]. For example, the

greening up and harvest of agricultural land multiple times throughout the year may help distinguish it from a young deciduous forest that looks spectrally and spatially identical at peak greenness or mid-winter, but only has a single green up and senescence each year. A new paradigm in accurately classifying long time series could simplify the previously discussed workflows. We predict that a model which automatically extracts representative non-linear spatial-spectral-temporal relationships that are embedded in year-long time series of multi-spectral remote sensing data may be more robust and generalizable across time. A recurrent convolutional neural network that incorporates spatial and temporal contexts may provide this capability.

### 1.2. Deep Learning

Deep learning methods are a powerful technique for automatically learning non-linear relationships and extracting semantically meaningful features from spatial and sequential data [29]. Prior studies have often shown deep learning methods surpass conventional methods (e.g., random forest, support vector machine) in many remote sensing problems [30], thus there is considerable motivation to bring deep learning into the field's mainstream (see [31] for a succinct case study and [32] for a comprehensive review). Convolutional neural networks (CNNs) and recurrent neural networks (RNNs) have been responsible for the majority of recent advances in artificial intelligence, driving major successes in computer vision [33–35] and sequence analysis [36,37] respectively.

CNNs ingest data in multi-dimensional arrays (e.g., images, video) and apply a series of learned filters that transform the raw data into higher level features that represent the original input data through layers of increasing abstraction. In remote sensing, this effectively means CNNs learn a spatial or landscape context (see [38] for a conceptual overview). These models excel at remote sensing tasks such as scene recognition, object detection, and image segmentation [32]. Examples include mapping intertidal habitat from drones [39], identifying tree crowns from airborne imagery [40], segmenting buildings in high resolution satellite imagery [41], and mapping land cover from satellite [42]. Though generally CNNs are used for extracting spatial patterns they can also be used for time series classification [43] and have successfully be applied to remote sensing time series by turning the 2D convolutions into 1D temporal convolutions [44].

While CNNs typically learn spatial patterns, RNNs were designed to learn temporal or sequential relationships among data points. They are currently the basis of many state-of-the-art approaches to natural language processing, stock market price prediction, and robotic control [36,37]. RNNs have been used frequently within remote sensing to learn sequential patterns in the spectral bands of hyperspectral data rather than in time [45], and a broad array of work has demonstrated the promise of RNNs when there is a temporal dimension to remote sensing data [46–48].

Combining these two architectures into a recurrent convolutional neural network (RCNN, sometimes called convRNN or ConvLSTM) is a novel and promising approach for extracting the complex relationships between space and time necessary for highly accurate and temporally generalizable land cover predictions [49]. Much less frequently used in remote sensing, RCNNs have become state-of-the-art in problems with joint spatial and sequential dimensions such as precipitation predictions [50], agriculture classification [24], and forest classification and structure regression [28].

While the bulk of deep learning applications in change detection over the past five years has been using CNNs and RNNs there is a growing set of new models and approaches (reviewed by [51] along with an excellent discussion of implementation). Multi-sensor fusion within deep learning algorithms is promising and fusing optical and radar data streams has been shown to better discriminate among land cover classes [52]. New models such as Fully Convolution dense Dilated Networks, which are finding success in other vision problems, may help in remote sensing problems where image attributes across a large range of scales are important for classification and receptive fields need to be large [53]. Optimizations in loss functions should also be leveraged that help mitigate

severe class imbalance as is the case in many land cover classifications [53,54]. A new type of mechanism called attention allows a model to look at an input sequence and decide at each step which features from other time steps are important in a time series. Attention layers within deep learning models applied to remote sensing data are proving to be highly effective at classifying time series and more robustly handling noisy data [55,56]. Transformer models, the primary model type utilizing attention, have been extraordinarily successful in natural language processing and have recently begun to impact the field of computer vision where they could help resolve many longstanding remote sensing problems in classification, object detection, segmentation, time series analysis, and even the integrating of remote sensing data with other modalities such as text and vector data [57]. We will focus here on CNNs, RNNs, and RCNNs but many new models will likely be important in fully resolving the problem of accurate land cover predictions across time and into periods before proper validation data exist.

One particularly valuable aspect of deep learning in land cover mapping is improved generalizability compared to traditional machine learning approaches. Many traditional classifiers are specific to a single satellite image or time period and thus only learn a fragile representation of the statistical characteristics of the classes of interest [14,31]. The increased generalizability of deep learning methods can be attributed to their learning of hierarchical abstract feature representations from the raw input that provide better separation between classes than traditional methods, making them less sensitive to slight differences in sensor characteristics or noise [29,32]. In an extreme example, a deep learning model trained to detect change on images of Mars was still able to extract meaningful features for change detection when applied to images of Earth and the Moon [58].

Explicitly accounting for phenological patterns and intra-annual variation, particularly in agricultural areas, is a complex task, especially given shifting climate conditions. Thus, if annual classification outputs are acceptable for a project and sufficient imagery across each year is available, a deep learning model that learns the patterns indicative of each class by ingesting an annual stack of multispectral imagery may be an ideal approach to correctly predict land cover across time. It is our conclusion that a model capable of learning a temporally generalizable representation of each land cover class of interest could be a reproducible, rapidly deployable, and highly accurate solution to many of these issues.

### 1.3. Objectives

In this study, we (1) developed an RCNN for predicting land cover class, (2) evaluated this model across time and space and benchmarked it with conventional machine learning methods and common deep learning approaches, and (3) applied this model to predict land cover across two decades in order to analyze change in the coastal plain of North Carolina, USA.

## 2. Materials and Methods

### 2.1. Study Area

The coastal plain of North Carolina (Figure 1) is a series of low relief drowned river valleys with a chain of barrier islands enclosing the nation's second largest estuary. While there are high levels of development along many barrier islands, the population density is generally low across the coastal plain. This region includes vast tracts of farmland, timberland, natural forest, and wetlands that sit only centimeters above sea level. Hurricanes and SLR are driving saltwater intrusion which kills forests and leaves vast zones of bare and fallen trees in their wake [59]. These aptly named ghost forests have spread out along the region's fractaline river networks and coastlines (Figure 2). Fringing emergent wetlands then migrate into these ghost forests [60]. Human-made channels once meant to drain the land for farming have exacerbated SLR in this region, as these channels turn into conduits for increased saltwater intrusion [10]. In areas where the saltwater intrusion is less intense, SLR is still causing farms to be abandoned as they become too wet and turn into swamp

forests (Figure 2). Not constrained to the coastal plain of North Carolina, similar transitions are occurring in many low lying coastal regions across the world [61].

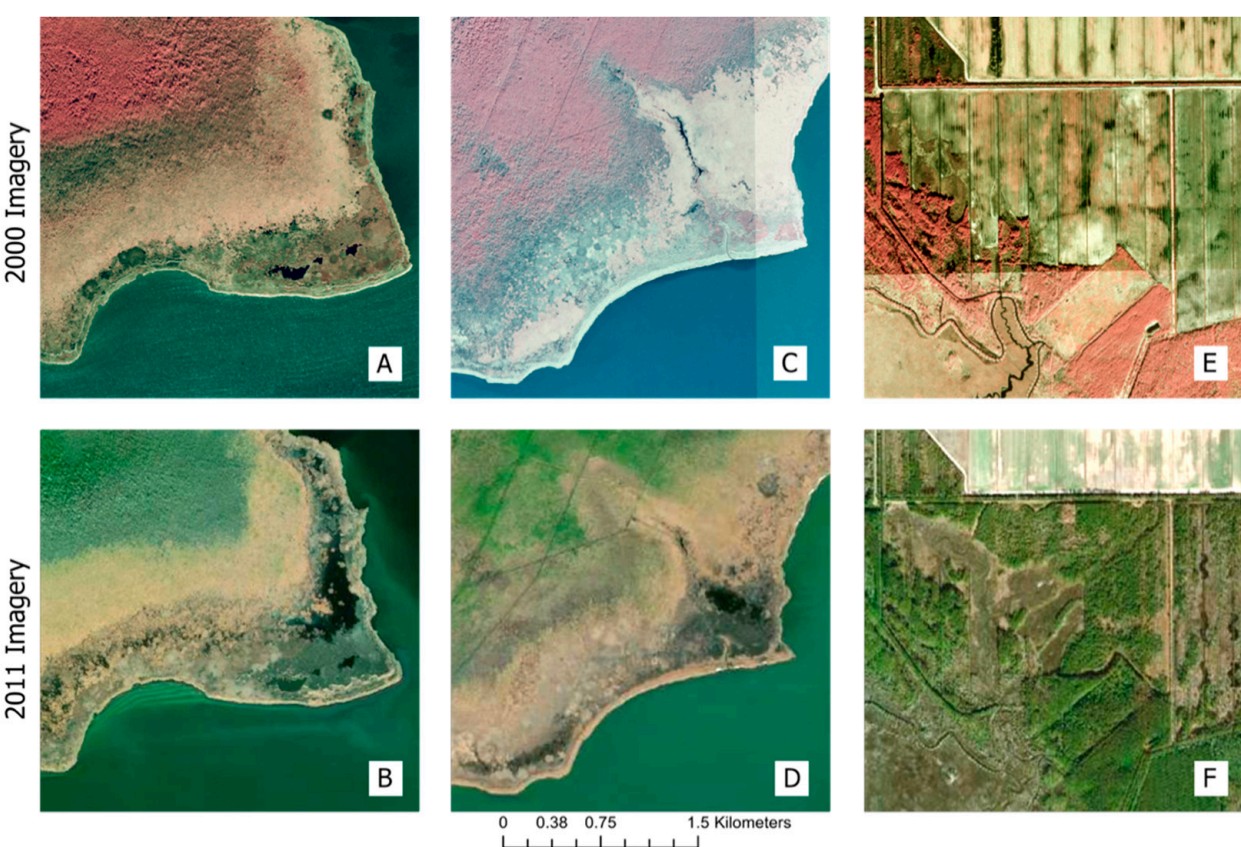

**Figure 2.** Comparisons of areas along the coastal plain in high resolution imagery from 2000 and 2011. Note that 2000 imagery is shown in false color (NIR-R-G) and 2011 imagery is in true color. Panels (**A–D**) show the encroachment of wetland into forested areas. Panels (**E,F**) show what was initially agriculture turning into a wetland forest.

Natural vegetation in this study is classified as either "forest" or "wetland", where forest is dominated by woody species (i.e., trees) and wetland is dominated by emergent herbaceous plants including both freshwater/riverine and salt marsh/estuarine types. An important distinction is that much of the forested land on the coastal plain, particularly at low elevation, exhibits wetland hydrology. These forested wetlands, including pocosins, palustrine forests, and bald cypress swamps, are particularly important ecosystems, sequestering more carbon in biomass than their herbaceous wetland counterparts [62]. For ease of classification, forested wetlands are included in the forest class, but it is these low elevation forested wetlands which experience natural inundation cycles that are most vulnerable to SLR and saltwater intrusion and facilitate the encroachment of herbaceous wetland vegetation as forests retreat [63].

### 2.2. Data

#### 2.2.1. Satellite Imagery

We used the Landsat 5 analysis ready data (ARD) surface reflectance product for this study [64] (Figure 3). Landsat 5 ARD is corrected to surface reflectance using the Landsat Ecosystem Disturbance Adaptive Processing System (LEDAPS) Surface Reflectance Algorithm [65]. Full technical details can be found at [66]. There are known issues with the cloud masking algorithm used in this product, particularly around coastal areas where bright sand and intertidal areas are often misidentified as clouds. We did not use the cloud mask for this study because these false positives are significant in our study area.

Instead, we manually filtered and visually verified images that were cloud free. Images with optically thin clouds, small amounts of smoke (e.g., from wildfires or controlled burns), or clouds over the water were included. This was both a necessity to reach five images per year and we assumed that with multiple time steps in a single year the models could become insensitive to a single timestep with minor clouds. Nineteen eighty-nine was the earliest year with five images relatively evenly spaced throughout the year and meeting our cloud filters and 2011 was the latest year meeting these requirements. Normal operations with Landsat 5′s Thematic Mapper instrument ended in November 2011 and the satellite was decommissioned totally in 2013. Two thousand was the earliest year meeting these requirements that also had sufficient high-resolution data for testing model accuracy. All image dates and tile IDs can be seen in Table 1.

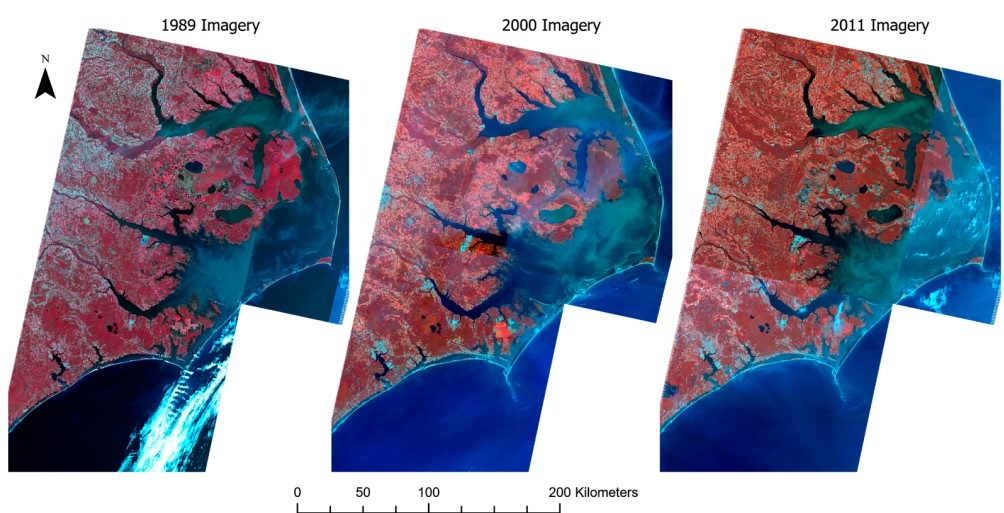

**Figure 3.** Examples of Landsat 5 imagery used for this study from all three time periods visualized in false color. Note that clouds over water and optically thin clouds over land were included. From left to right: April 1989 imagery, August 2000 imagery, and August 2011 imagery.

**Table 1.** Dates for all Landsat images used organized by tile ID and year. Date format is YYYY-MM-DD. Tile 028011 is the northwest component of the study area, tile 029012 is the furthest east primarily encompassing the barrier islands of North Carolina, and tile 028012 is the southern section of the study area.

|  | Date Order | 1989 | 2000 | 2011 |
|---|---|---|---|---|
| Tile 028012 | 1 | 1988-12-05 | 2000-01-05 | 2011-01-03 |
|  | 2 | 1989-03-11 | 2000-04-10 | 2011-03-08 |
|  | 3 | 1989-04-12 | 2000-08-16 | 2011-07-30 |
|  | 4 | 1989-10-05 | 2000-10-03 | 2011-08-31 |
|  | 5 | 1989-10-21 | 2000-10-19 | 2011-11-03 |
| Tile 029011 | 1 | 1989-03-11 | 2000-02-22 | 2011-01-03 |
|  | 2 | 1989-04-28 | 2000-04-10 | 2011-03-08 |
|  | 3 | 1989-06-15 | 2000-08-16 | 2011-07-30 |
|  | 4 | 1989-10-05 | 2000-10-03 | 2011-08-31 |
|  | 5 | 1989-10-21 | 2000-10-19 | 2011-10-18 |
| Tile 028011 | 1 | 1989-03-11 | 2000-01-21 | 2011-01-03 |
|  | 2 | 1989-04-12 | 2000-04-10 | 2011-03-08 |
|  | 3 | 1989-06-15 | 2000-05-12 | 2011-08-31 |
|  | 4 | 1989-10-05 | 2000-08-16 | 2011-10-18 |
|  | 5 | 1989-10-21 | 2000-10-19 | 2011-11-03 |

The ARD output is necessary because consistent pre-processing with the best possible algorithms and consistent reflectance values is critical to analyze change through time.

Considerable intra-annual coverage is available with Landsat 5 ARD. There are an average of 15 non-cloudy acquisitions per year for pixels contained within our study region over the lifetime of Landsat 5, providing high temporal density at first pass [67]. This is much more limited given our requirements for the whole tile to be generally cloud-free. The consistency of these products for long-term time series analysis has been demonstrated, though there is room for continued improvements such as better cloud masking and bidirectional reflectance distribution function corrections [68].

Data for the model consisted of Landsat 5 ARD at five time steps (winter, spring, summer, early fall, late fall) using bands 1, 2, 3, 4, 5, and 7. Band 6, Landsat 5′s thermal band, was not used. We calculated the normalized difference vegetation index (NDVI) as a 7th band. NDVI is defined as:

$$NDVI = \frac{B4 - B3}{B4 + B3} \tag{1}$$

where *B*4 and *B*3 are the near-infrared and red Landsat 5 bands, respectively. Figure 4 shows the annual mean of each band and NDVI in the six land cover classes, along with standard deviation. The large standard deviation and overlap between classes hints at the challenge of differentiating classes based on spectral properties alone.

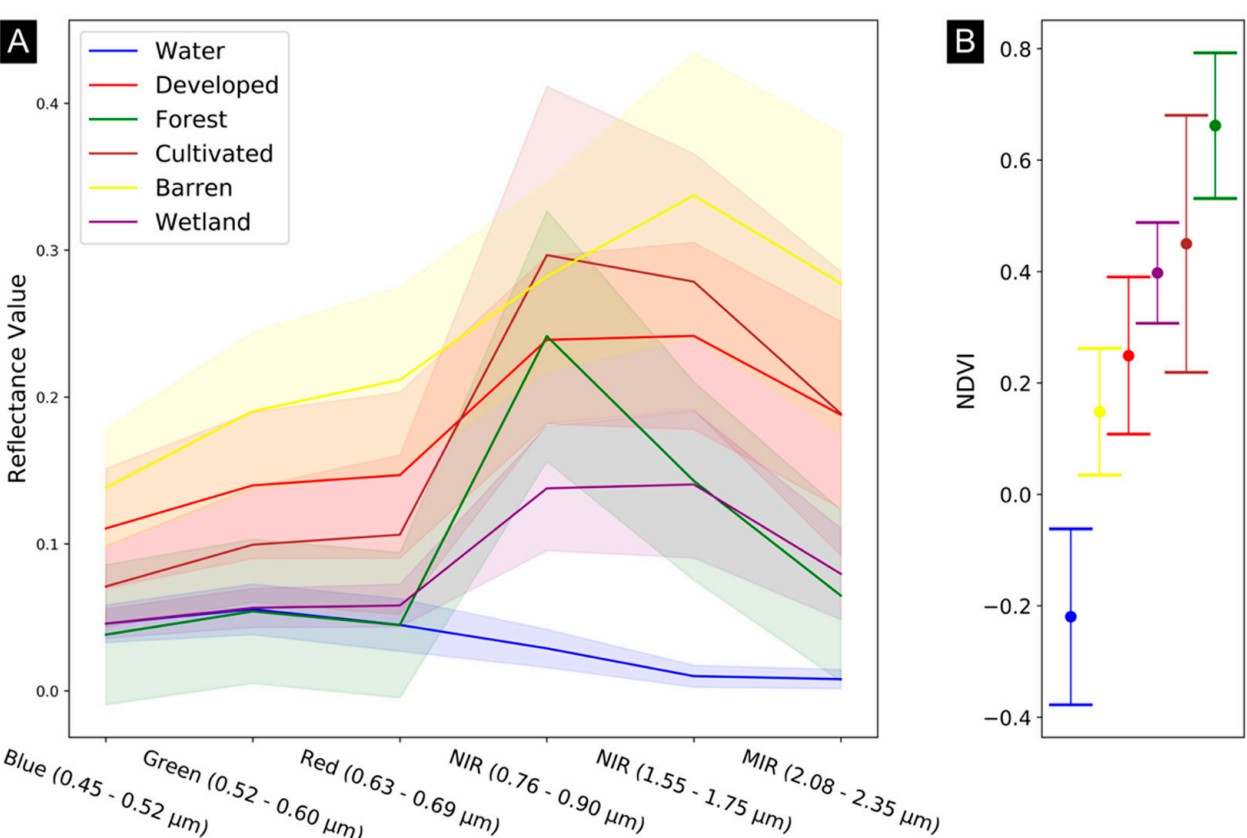

**Figure 4.** Spectral signatures and NDVI values for each class in the study. In panel (**A**) the heavy lines represent the mean for each class and the shaded area is the standard deviation. On panel (**B**) points represent mean NDVI and bars represent the standard deviation. Values were based on training labels and reflectance values from August 2011.

We normalized the observations by calculating the mean and standard deviation of each band for 200,000 random pixels across the study site from each time step of the 2011 data and then subtracting each pixel by that mean and dividing by the standard deviation. The same normalization values and calculation were applied to 2000 and 1989 data.

### 2.2.2. Training, Validation, and Test Data

Our approach for analyzing temporal generalizability was to train and validate models on data from 2011 and choose the version of each model that performed best on the validation data, then to test it on unseen data from 2011 and 2000 (Table 2). We assessed both overall accuracy and the change in accuracy when classifying data from a decade earlier with slightly different time steps for the five Landsat images.

**Table 2.** Sample count and validation process for each dataset used in model development. The training dataset was used to train all models and the validation dataset was used to choose the best model iteration from training. The two test datasets were used to assess the output classification on previously unseen data from the year it was trained (2011) and a decade earlier (2000). The $3 \times 3$ homogeneity filter for the training data ensured all pixels neighboring the pixel going into the model were the same class, increasing the chance that the pixel was representative of the intended land cover without the need to manually validate tens of thousands of pixels. Validation and train datasets were manually validated with visual inspection.

| Dataset | Sample Count (per Class) | Filter Process |
|---|---|---|
| 2011 Train | 1500 | Homogeneity filter only |
| 2011 Validation | 120 | Manual validation |
| 2011 Test | 120 | Manual validation |
| 2000 Test | 100 | Manual validation |

Class labels for model development were based on a merged and filtered version of the 2011 NLCD [69]. The NLCD class accuracies are often below 80% and very few of their change products have over 70% accuracy [70], with no outputs before 2001. This combination of temporal range and accuracy make the dataset unsuitable for analyzing subtle climate induced changes as in this work, but they can be used as the basis for labels to train a regional model. We merged classes from the NLCD into water, developed, forest, agriculture, wetland, and barren categories. Developed in this study refers to low, medium, and high intensity areas of development.

We then created training, validation, and test datasets by randomly selecting pixels, stratified by class, across all three Landsat tiles. These pixels are used as the center of $9 \times 9$ pixel patches. We first generated testing data, ensuring each center pixel was more than 9 pixels apart, giving a margin of 1 pixel between patches (Figure S1). We then generated validation data, ensuring it did not overlap spatially with the test data or itself. Finally, we generated training data, relaxing the constraint that center pixels be 9 pixels apart from other training data, but not test or validation. This guaranteed that examples were not spatially overlapping and had a margin of at least 1 pixel between each patch. Testing and validation data were manually validated, and training data was not. The training data was filtered using a $3 \times 3$ homogeneity filter to ensure pixels were only coming from a region of consistent land cover. The objective of this filter was to eliminate edge cases and train the model with more representative examples of each land cover type without the need to manually validate tens of thousands of pixels.

We performed manual validation in ArcGIS Pro using high resolution aerial imagery (<0.3 m resolution) from both 2011 and 2000. Imagery for validating the 2011 validation and test data was acquired from the ArcGIS WayBack Machine and added to ArcGIS Pro using the Add Data from Path tool and this URL https://www.arcgis.com/home/item.html?id=903f0abe9c3b452dafe1ca5b8dd858b9 (accessed on 10 May 2020). Imagery for manual validation of 2000 test data was acquired using the USGS EarthExplorer and consisted entirely of a USGS survey from 2000 available as a Digital Orthophoto Quadrangle (DOQ). During manual validation, we also used the summer Landsat tile from each year, visualized as NIR-G-B, to ensure no visible change had occurred in the months between the high-resolution surveys and the midpoint of the year. Pixels comprising exactly half of two land cover types were rejected. This resulted in 720 pixels (120 per class) for the 2011 validation

dataset, 720 pixels (120 per class) for the 2011 test dataset, and 600 pixels (100 per class) for the 2000 test dataset. Manual validation took approximately five hours for each of the 2011 validation, 2011 test, and 2000 test datasets. This visual validation process has been shown to be effective for model validation [71], permits a much larger dataset than could be attained by ground-based methods, and is often the only viable option for creating labeled data for decades past. While the 9000 training pixels were not individually validated, 150 pixels of each class were validated using the same process as above to assess overall accuracy of the filtered and merged NLCD data. The overall accuracy of this subset of filtered and merged training data was found to be 93.5%.

### 2.2.3. LiDAR Data

Light detection and ranging (LiDAR) data from the NOAA Office of Coastal Management collected for NOAA's sea level rise analysis was not used as a model input, but was used as an ancillary dataset to provide ecological context for changing land cover classes and to mask out deep water in the study area. This dataset has a horizontal resolution of 5 m, a vertical resolution of 9 cm, and was referenced to the North American Vertical Datum of 1988 (NAVD88). This higher resolution LiDAR data was downscaled to match Landsat 5 resolution and pixel locations.

### *2.3. Classification Methods*

We tested several commonly used classification methods across the study area and compared their performance for temporal generalization. We compared random forests (RF) and support vector machines (SVM) as well as a multi-layer perceptron (MLP), a CNN, an RNN, and an RCNN. To better understand influences on temporal generalization, we also tested a random forest with only a single pixel as input (RF_1) and an RCNN concatenated with the RNN (RCNN_LSTM). The idea behind this was to test if additional branches of the network (sensu [28]) were able to better incorporate the temporal dimension and increase generalizability. We performed an additional experiment to assess the benefit of using multiple time steps compared to imagery from a single date by only using the late spring date in the RCNN input (RCNN_SINGLE) this RCNN had the same structure the standard RCNN but with a different input shape described below. All methods output discrete land cover classifications for each Landsat pixel.

The RF and SVM models were implemented with the python module scikit-learn [72] with all default settings except for increasing the number of estimators within the RF algorithm to 500. All deep learning models were implemented with the python module keras [73]. All models used rectified linear units as activation functions and end in a dense layer of 6 units with a softmax activation. The MLP was composed of three dense layers interspersed with 20% dropout layers for a total of 92,006 parameters. The CNN was composed of three convolutional and max pooling layers, dense layers of 256 units, 64 units, and 32 units, and interspersed with 20% dropout layers for a total of 121,574 parameters. The RNN is composed of nine recurrent layers and a dense layer of 64 units before the output for a total of 90,182 parameters (Table S1). The recurrent components of the RNN were densely connected long short-term memory (LSTM) layers. Densely connected means that the output of every layer is concatenated with the input of subsequent layers. Previous work has demonstrated that these densely connected blocks speed up convergence and improve accuracy [74], though they increase the number of weights that must be optimized in the network. The RCNN is composed of two convolutional LSTM blocks, a max pooling layer, then flattened before a dense layer of 64 units before the output layer for a total of 496,326 parameters (Table 3). The RCNN_LSTM is the RCNN and the RNN with features concatenated together before the final 64 unit dense layer.

**Table 3.** Architecture for recurrent convolutional neural network as output as printed from the model.summary method in keras showing the layer name, the shape of that tensor, and the number of trainable parameters in each layer. The tile_input layer shows the input of 5 time steps, by 9 pixels, by 9 pixels, by 7 image bands. This input is put through two Conv2DLSTM layers, followed by a $3 \times 3$ max pooling. The output from the max pooling layers is flattened, run through a dense layer, and output into the six final classes.

| Layer (Type) | Output Shape | Param |
|---|---|---|
| tile_input (InputLayer) | (None, 5, 9, 9, 7) | 0 |
| conv_lst_m2d_1 (ConvLSTM2D) | (None, 5, 7, 7, 64) | 163,840 |
| conv_lst_m2d_2 (ConvLSTM2D) | (None, 5, 5, 64) | 295,168 |
| max_pooling2d_1 | (MaxPooling2 (None, 3, 3, 64) | 0 |
| flatten_1 (Flatten) | (None, 576) | 0 |
| dense_1 (Dense) | (None, 64) | 36,928 |
| landcover (Dense) | (None, 6) | 390 |

The input to the RCNN model was a $9 \times 9$ tile of pixels with five time steps and 7 bands (i.e., $x_i \in \mathbb{R}^{9 \times 9 \times 5 \times 7}$), and the output was the predicted land cover class of the center pixel (Table 3). Input for the RCNN_SINGLE model was a $9 \times 9$ tile with one time step and 7 bands (i.e., $x_i \in \mathbb{R}^{9 \times 9 \times 7}$). Since the RF, SVM, and MLP require vector inputs, we flattened the input into an array with 2835 elements for these methods ($x_i \in \mathbb{R}^{1 \times 2835}$). Similarly, the RF_1 model input has only one pixel ($x_i \in \mathbb{R}^{1 \times 35}$). Since the CNN requires 3-dimensional inputs, we flattened the spectral and temporal dimensions ($x_i \in \mathbb{R}^{9 \times 9 \times 35}$). The RNN requires sequential (not spatial) inputs, thus we only used the center pixel as input ($x_i \in \mathbb{R}^{5 \times 7}$).

The RF and SVM models were fit on the training data using the standard scikit-learn fit function. All deep learning models were trained for 60 epochs using a batch size of 25. The model weights in the epoch where the model had highest accuracy on the 2011 data was saved for testing.

Since the training dataset was not manually validated and had an accuracy of 93.5% there are errors that cause noise in the labels during training. An approach called fine-tuning, in which a model is first trained with a large dataset of noisy labels (or examples from another domain) and then trained with a smaller dataset of more relevant or higher quality examples, has been shown to improve model performance in previous work [75,76]. To assess the benefit of fine-tuning the RCNN with higher quality labels, we used the aforementioned subset of 900 manually validated pixels (150 per class) from the training dataset to train the RCNN after the standard training procedure (RCNN_FINE). This dataset was only used in the RCNN_FINE model.

## 3. Results

### 3.1. Model Results

Model testing shows that all models are capable of high accuracy across all classes on the 2011 data they were trained on, but the RCNN (Table 3) had higher accuracy than other models when predicting on the 2000 data (Figure 5). Based on this result it was used to classify the 1989 data for our change analysis. Model accuracy in 2011 was not always a good indicator of temporal generalizability. This may indicate that some models are good at predicting on data but the relationships used to make these predictions are brittle across time. In general, all models had difficulty distinguishing between cultivated vs. forest and developed vs. barren classes, and to a lesser extent the wetland vs. cultivated classes as shown in the more detailed confusion matrices from the top performing RCNN model (Figure 6). This is expected because this region contains substantial timberland which is spectrally and spatially similar to cultivated for the first year after harvesting. Developed is spectrally indistinguishable from barren (e.g., sand and concrete parking lots) and only

spatially distinct, and many farms are verging on transitioning to wetland. As anticipated, the RCNN model with only a single time step was considerably worse, which suggests that the RCNN is able to extract meaningful patterns out of the phenological data which may help it ignore ecological or sensor irregularities that only exist in a single time step. Fine-tuning the RCNN on a dataset of 900 manually labeled pixels (150 per class) from 2011 did not improve model performance (Figure 5).

### 3.2. Ecological Results

The comparison of 1989 land cover to 2011 land cover based on the RCNN model output shows striking change in two decades (Figure 7 and Table 4). Changes occurred primarily within the forest, wetland, and cultivated classes. Across the study site, low-lying farms and forests transitioned into wetland (Figures 7 and 8) causing the total area of pixels classified as wetland to more than double from 719 km$^2$ to 1520 km$^2$ (Table 4). This change was most common along water boundaries and in areas below 0.5 m elevation (Figure S2). A small portion of wetlands have transitioned to forest, 62 km$^2$, and an even smaller portion of wetlands have transitioned to open water, only 14 km$^2$ (Table 4).

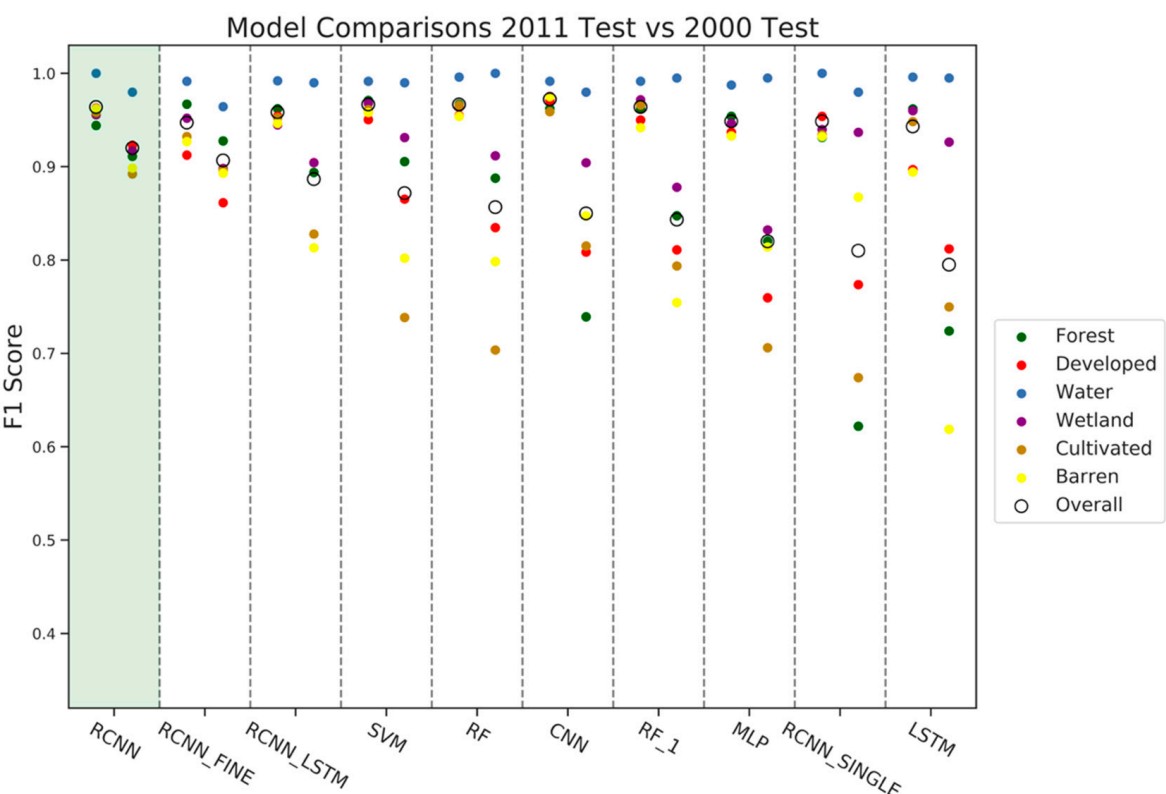

**Figure 5.** Comparison of model accuracy ordered from left to right by overall accuracy from the 2000 test set. F1 score is shown for each class. F1 score is the harmonic mean of producer's and user's accuracy. The left side of each model's column is the 2011 test data and the right side is the 2000 test data. Points are colored by class.

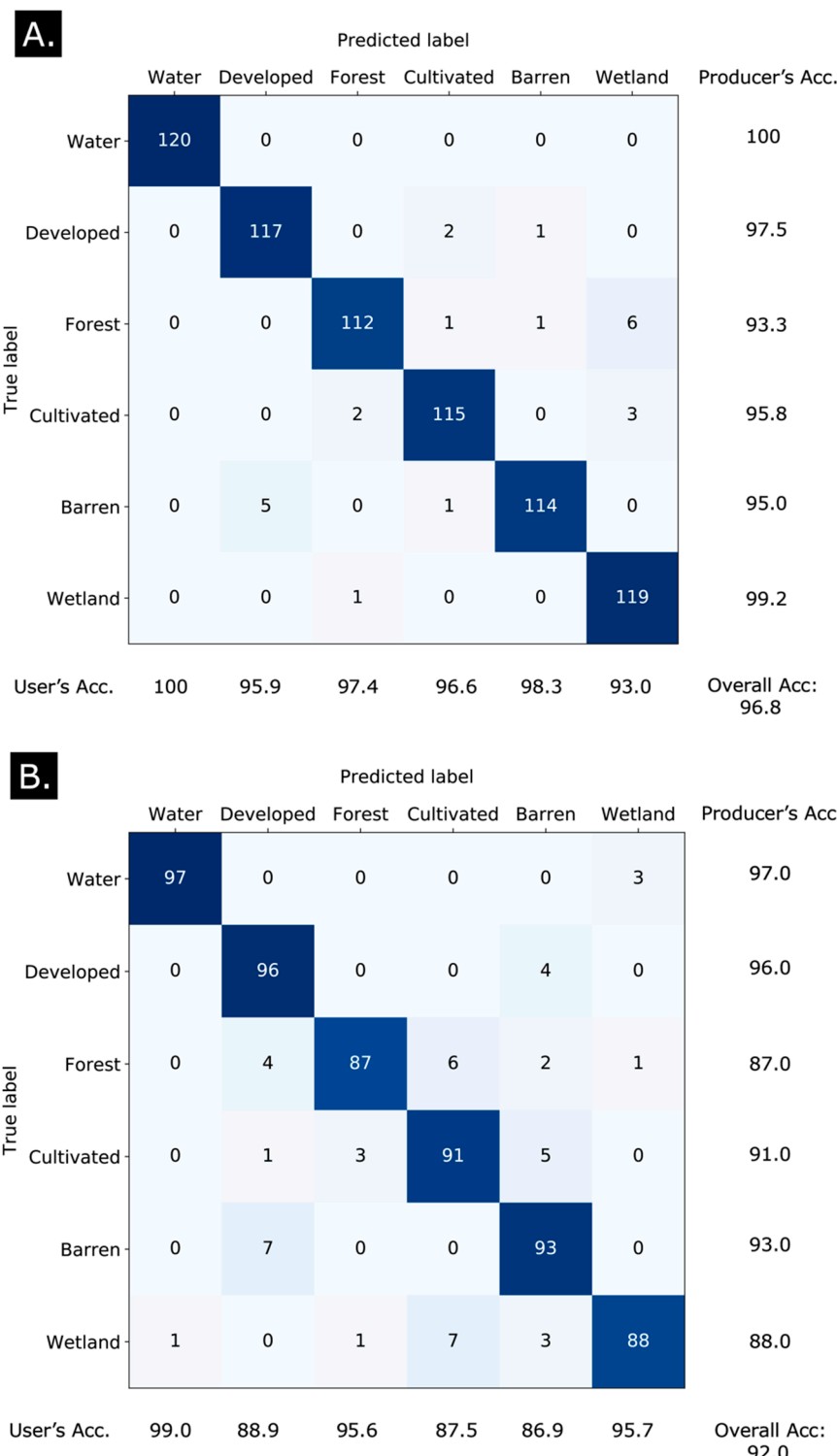

**Figure 6.** Confusion matrix including user's, producer's, and overall accuracy for the recurrent convolutional neural network on (**A**) 2011 test data and (**B**) 2000 test data. The difference in per class totals (120 vs. 100) is based on the process of generating 150 pixels per class from the NLCD dataset and then manually validating. The correct class was visually validated, mixed pixels were rejected, and then all classes were subset to the same number as the class with the least number of validation pixels.

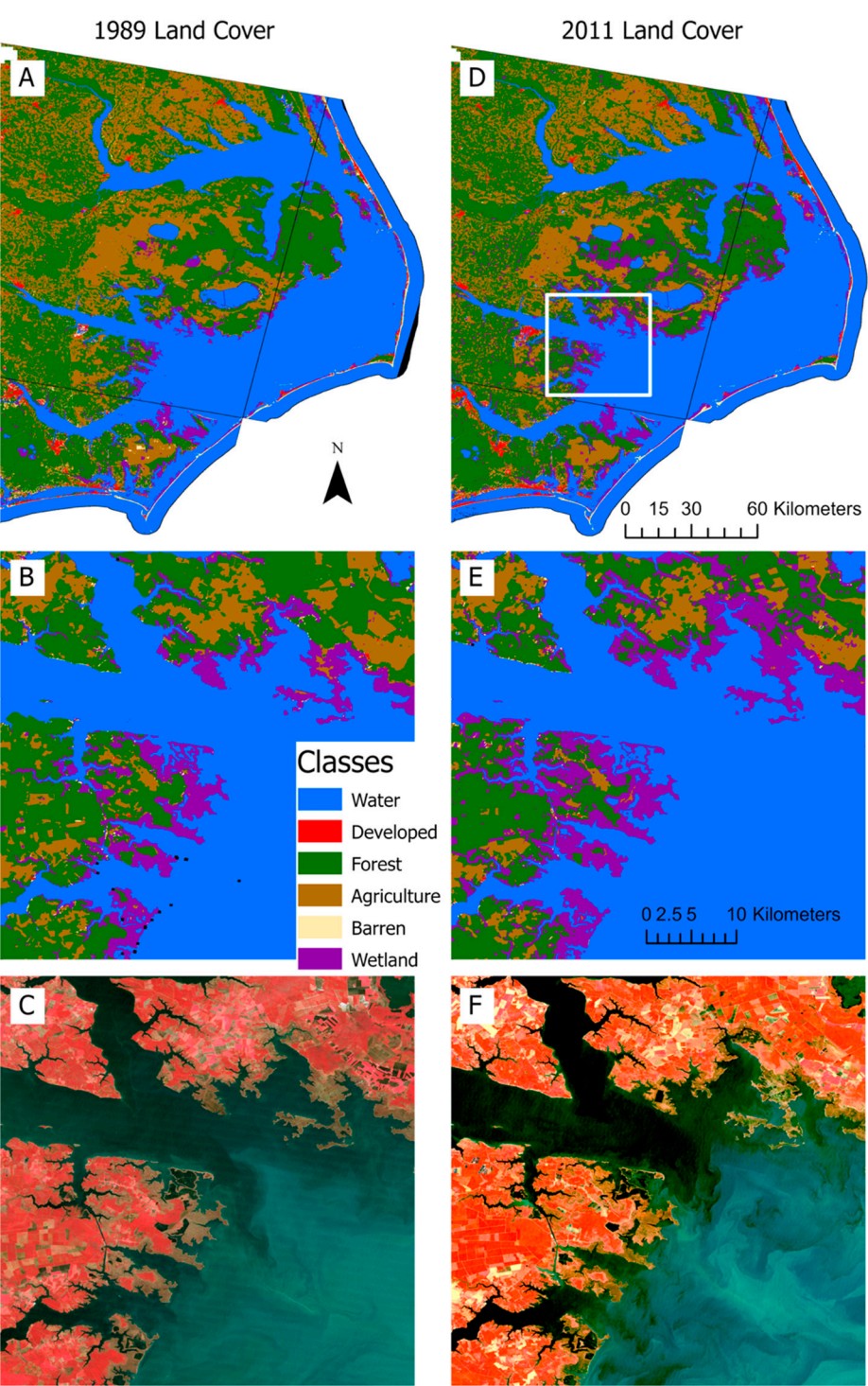

**Figure 7.** Final output land cover classes from the recurrent convolutional neural network in 1989 (panel **A**) and 2011 (panel **D**). The zoomed region in panels (**B**,**E**), (**C**,**F**) is denoted by the white rectangle in panel (**D**). Panel (**E**) shows the creep of wetland inland. Panel (**E**) also shows the large number of pixels that changed to wetland and appear to be along field boundaries as well as ditches and canals that provide connectivity to saltwater. Panels (**C**,**F**) show Landsat imagery from the spring of the 1989 and 2011, respectively, visualized in false color as NIR-R-G for comparison with model output.

**Table 4.** All transitions in km² between land cover classes and the total percent change of each class. Rows add up to the total amount of each class in 2011 and columns show how that class was classified in 1989.

|  | Water 1989 | Developed 1989 | Forest 1989 | Farm 1989 | Barren 1989 | Wetland 1989 | 2011 Totals | Percent Change |
|---|---|---|---|---|---|---|---|---|
| Water 2011 | 459 | 1 | 5 | 10 | 8 | 14 | 498 | −0.02% |
| Developed 2011 | 3 | 203 | 52 | 52 | 43 | 2 | 355 | 3.87% |
| Forest 2011 | 3 | 25 | 8079 | 1009 | 58 | 62 | 9236 | −1.95% |
| Farm 2011 | 1 | 79 | 582 | 3936 | 93 | 16 | 4707 | −9.79% |
| Barren 2011 | 15 | 21 | 41 | 38 | 116 | 8 | 239 | −33.43% |
| Wetland 2011 | 18 | 12 | 660 | 173 | 42 | 617 | 1520 | 111.40% |
| 1989 Totals | 498 | 341 | 9419 | 5218 | 359 | 719 | 16,554 | |

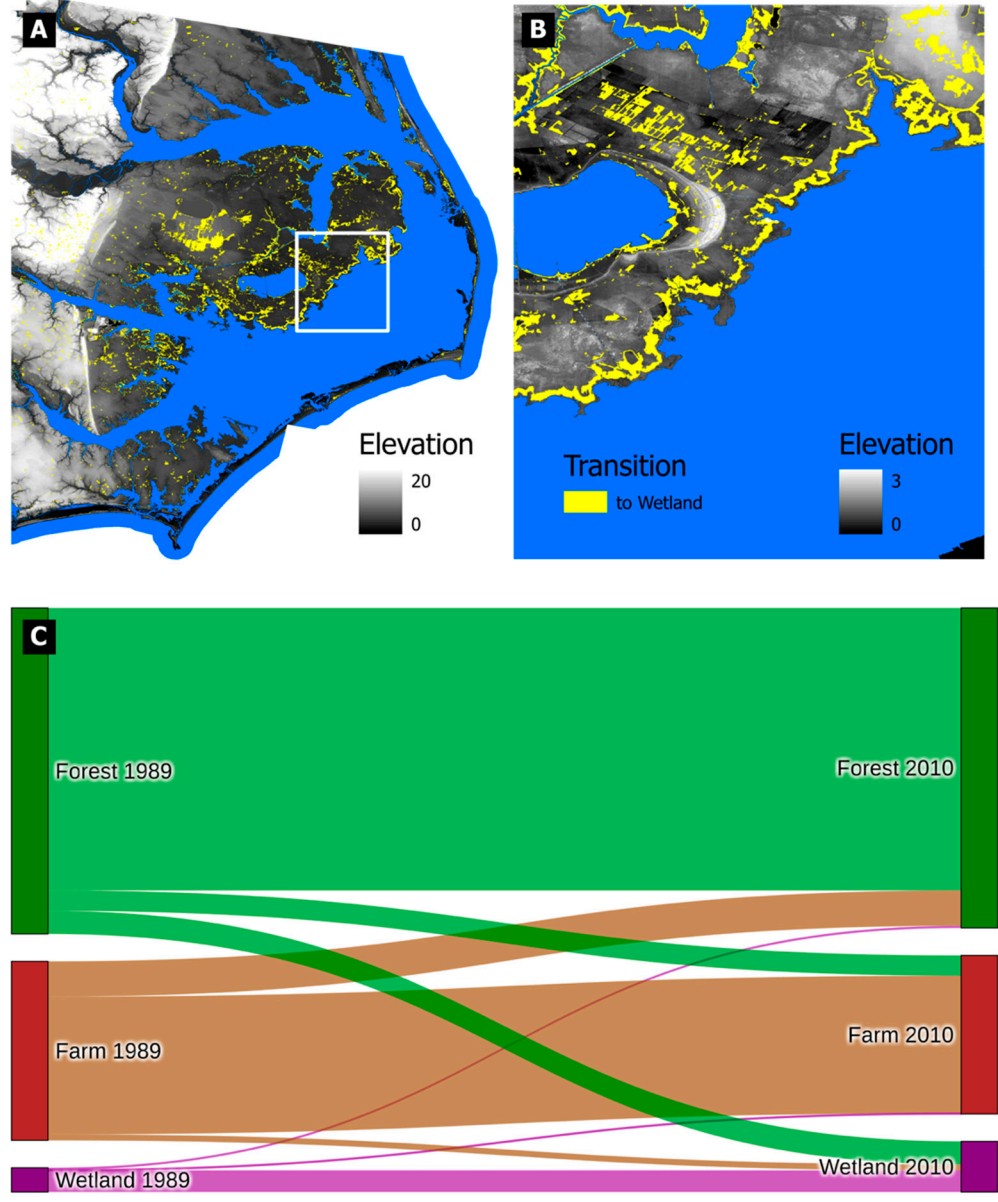

**Figure 8.** Panels (**A**,**B**) show in yellow the pixels that were classified as forest or agriculture in 1989 but as wetland in 2011 overlaid on a map of elevation. Panel (**C**) shows the quantity of pixels that transitioned between forest, agriculture, and wetland from 1989 to 2011.

The total area of pixels classified as forest decreased from 9419 km² to 9236 km² from 1989 to 2011. Approximately 660 km² of forest became wetland and 582 km² of

forest became cultivated land. However, the net change in forest cover is tempered by a significant amount of cultivated land transitioning to forest (1009 km$^2$). The transitions between cultivated land and forest were spatially distributed throughout the study area. The distribution of elevations where a transition between these classes occurred is similar to that of the whole study site. The area of pixels classified as developed increased minimally from 341 km$^2$ to 355 km$^2$.

## 4. Discussion

### 4.1. Implications of Ecological Change

While the loss of land to open water often garners major attention, it represents only a fraction of the overall change found to be occurring on the landscape. In contrast, the conversion of land between vegetation types (i.e., forest to wetland) is much more prevalent and yet has received little attention in the literature despite numerous ecological consequences.

The implications of a major transition of farmland and forested area into emergent wetland and the associated ecological shifts are manifold. We see an almost ubiquitous expansion of wetlands into coastal forests along the margins of the estuaries, resulting in a two-fold increase in wetland area from 1989 to 2011 (Figure 8). Many agricultural areas transitioning into wetland are in proximity to large artificial channels such as the Atlantic Intracoastal Waterway and drainage ditches that were commonly dug over the last century in this area [77] (Figure 7). Manda et al. [78,79] demonstrated the role that artificial channels play in driving salinization of wetlands in this region and this change was outlined in recent land cover mapping work [59]. The conversion of agricultural land to wetland is consistent with an ongoing trend of farm abandonment in coastal areas in response to field salinization and the rising costs of active drainage [80]. These changes are indicative of increased flooding and likely salinization of these areas from SLR.

The sea level trend from one of NOAA's nearby tide gauges (8652587 Oregon Inlet Marina, NOAA CO-OPS) is 5 mm/year, which is among the highest SLR rates along the US East Coast. Coupled with the extremely low elevation gradient (Figure 7), the forests of Eastern North Carolina are highly susceptible to the intrusion of saltwater. This follows and further quantifies observations of the transition of swamp forests into emergent marsh along North Carolina estuaries [10,81]. Regression estimates of forest edge measurements from historical imagery suggest forest retreat rates in the last century (post-1875 sea level rise acceleration) have ranged from approximately 1.5 to 4.5 m/year [2,82], which would amount to 30–90 m (1–3 30 m Landsat pixels) of horizontal change, respectively, during the 1989–2011 time frame of our analysis. Considering that much of the inner coastline experienced forest-wetland shifts much greater than 90 m (3 pixels) (Figure 7), our results may indicate an acceleration of forest retreat in recent decades.

The fate of wetland areas in a world of rising seas is uncertain. Much work has been done to assess their vulnerability to SLR, balancing their ability to keep pace with SLR and maintain area with their relative sediment supply, topography and rate of upland migration [83,84]. Many salt marshes along the US coast are experiencing dramatic shoreline loss [82,85]. Within the study area, recorded marsh shoreline loss rates range from 0.24 m/year [86] to 0.75 m/year [87]. We do not see major changes of wetlands to water over this two-decade time period, either because the marsh shoreline is stable or, more likely, the amount of change is still within one pixel at the imagery resolution (30 m/pixel).

The implications of shifts between forest and agriculture are challenging to discern. This is largely because shifts from forest to agriculture may reflect harvesting of timberland, reverting it to a state where it is classified as agriculture, and shifts from agriculture to forest may indicate either normal growth of timberland or permanent change from agriculture to wetland forest.

The methods developed in the current study, and the results presented above, have important implications for efforts to restore coastal ecosystems and increase their resiliency in the face of global change. There is increasing interest in using nature-based solutions,

such as living shorelines [88], to dampen the effects of sea level rise and storms in coastal regions, and these efforts require an understanding of long-term trends in coastal ecosystem change as part of an integrated spatial planning process [89]. The results provided in the present study clearly illustrate that site choice and prioritization for restoration efforts, can benefit from remotely sensed (through both satellite and drone-based approaches—see [90] analyses of land cover change contexts in coastal systems [89].

*4.2. Data Issues, Labels, and Noise*

Our goal was to create a model that could be applied to time periods without training, validation, or test data. Towards this goal we show that the model generalizes reasonably well in a year that it was not trained on which was a decade removed from the data it was trained on. Based on that finding, and with the assumption that the Landsat 5 sensor responses and data distributions would be similar, we apply the model even further back to 1989. Given the caveats that the 1989 classification has not been validated, there is still some error in the model output, and mixed pixels were excluded from the test accuracy, these metrics should be seen as an upper baseline. It must be taken into consideration that the analysis of change across time is based on these assumptions and that there is still uncertainty in the classification.

A qualitative visual validation using the input Landsat imagery does suggest that model output is generally correct, and most transitions are ecologically reasonable. The transition of farmland and forest to wetland occurs in low lying areas with high connectivity to saltwater. There are very few isolated pixels which reflects the reality of the landscape. The increase in development along the barrier islands is well captured spatially. There is a suspect shift from developed to farmland (Table 4), though on further inspection this is typically around roads or sandy lots that later became farm. There is likely some error where pixels span an ecotone, particularly with roads that are generally subpixel, and where the model initially classified bright fallow farmland as developed pixels that were later correctly classified as farmland.

We continually found that proper data cleaning and structure was as important for the final accuracy as model architecture and specific hyperparameters. For example, including the cloud mask which erroneously masked out bright barren pixels led to very few barren examples within the training data. This prevented accurate classification of that class and required not using the cloud mask. Manually validating each Landsat tile for anomalies and sensor errors ensured data was clean. Inspecting distributions of the reflectance values for each band in each class helped to catch a distorted Landsat 5 image that likely would have introduced error into the models. These processing changes and quality checks were much more beneficial than efforts to tune hyperparameters, such as the number of filters in the RCNN's convolutional layers or output neurons in the dense layers. Though beneficial, these manual inspection steps are time intensive and could be prohibitively so when applying this methodology across large regions. In non-coastal areas where the cloud mask is less error-prone, or once the current cloud mask algorithm has been improved, these manual inspection steps could be less intensive and better automated. For the time being, and without these improved automated quality checks, the evaluation here should be regarded as an upper baseline and accuracy would likely decrease without manual image selection.

Even though we performed additional processing on the NLCD (including merging classes and filtering by homogeneity), there is still some error in the resulting land cover labels (93.5% accuracy, or 6.5% error). The RCNN effectively ignored noise in the training data to learn features that were important for accurately separating land cover classes. The simple homogeneity filter we used to reduce noise in NLCD labels could be used in other regions of the US and would considerably lower the burden of creating labeled data for training data-hungry deep learning models. An additional step that would better assess the model's spatial generalizability would be to generate training, validation, and testing

data from separate regions within a study area rather than intermingled. A good example of this can be found in [49].

Distribution shifts in class reflectance values were an initial concern because of the long span of this study and potential climate impacts in the classes. The model does not appear sensitive to timesteps being offset by up to two months and it was trained with somewhat noisy data (e.g., time steps with optically thin clouds or imperfect labels). Thus, we expect it to be robust to minor changes in phenology and shifting reflectance values that might result from two to three decades of climate change. While it was not tested here, we assume the model would lose accuracy, potentially substantially, if all five timesteps were from a single season or if multiple time steps include clouds over the same pixels. While most years have appropriately spaced and cloud free imagery, if the model were being applied operationally this requirement would prevent it from being applied successfully every year.

One aspect of CNNs responsible for high performance compared to other methods in computer vision is their capacity to extract semantically meaningful "abstract" features from imagery. This is easily apparent in high resolution natural images (e.g., [91]), but less obvious in coarse remote sensing imagery with small tile sizes such as in this study. In these images there is not as much 2D spatial context to learn from and much of the information is contained in the spectral and temporal relationships of a single pixel. We hypothesize this is why the RCNN is a substantial improvement over the CNN, but this may not be the case if using imagery with higher spatial resolution.

### 4.3. Class Separability

The fractaline nature of coastlines with numerous ecotones creates a continuum of land cover that is a challenge for medium resolution imagery. Even with high resolution aerial imagery it can be challenging to visually differentiate a forest exhibiting wetland hydrology from a terrestrial forest since many species (oak, maple, pine) are present in both ecosystems. Agricultural areas undergoing restoration to natural forested wetlands that are common in North Carolina present similar challenges. Augmenting high-resolution RGB imagery with Landsat bands and band indices (e.g., NDVI) led to higher confidence in manual validation data.

Discrete class boundaries for phenomena that have continuous transitions may also introduce noise in labels when model predictions are forced to choose one class. However, it is that continuum of change—the transition itself—that we are often most interested in. The ecotones are the most important zones of change in a coastal environment but are the most difficult to classify accurately. Even with the best possible land cover classifier, approaches such as multi-label classifiers and harmonic modeling will be necessary to complement discrete outputs.

In this study area where abandoned farms are transitioning into swamp forests and emergent wetlands, discrete class boundaries are challenging. There can be a long period of time when an area is on the boundary of multiple land cover types. Timberland is nearly identical spectrally and spatially to farmland for the first year of growth. Controlled burns and wildfires can cause a forest to appear as a wetland for two to three years. Large areas of featureless concrete (e.g., airports, large warehouses) are spectrally identical to sand. Water bodies that grow thick with algae can have an NDVI comparable to nearby wetlands. Improving these distinctions, whether for a human analyst or a model, may require LiDAR, synthetic aperture radar (SAR), or additional spectral bands that can provide height, density, soil moisture, and improved spectral distinction. However, using these products for long-term analyses is limited by the fact that many of these data types are only available starting in the early 2010s if at all. In this study, despite reasonably high overall accuracy, a more precise understanding of change is hindered by the roughly 10% error that results largely from the challenge of separating these classes.

### 4.4. Future Methodological Work

Future work will entail adding more frequent land cover classifications, increasing the time span of land cover outputs, and adding finer class distinctions. Adding length to this record will require testing model generalizability across different sensors. Incorporating Landsat 4 and Landsat 8 could nearly double the length of the analysis to almost 40 years. More frequent classifications will better capture ecosystem transitions and intermediate states, providing insight into whether disturbance events, namely hurricanes, cause transitions or if they are gradual. To expand spatially will require assessing the limits of spatial generalizability into different ecological regions where a major question will be whether multiple models trained regionally or a single continental scale model is better [92]. Discriminating between additional classes will likely require more accurate training data labels, denser temporal stacks via products such as the Harmonized Landsat Sentinel-2 Dataset [93], and exploitation of new models such as attention-based Transformers [55], and networks ingesting multi-modal data streams [52].

### 5. Conclusions

In this study we assessed a variety of models for temporal generalizability in predications of landcover. We found that a RCNN which extracts spatial-spectral-temporal patterns from multispectral satellite time series may help expand our capability to use the full temporal extent of satellite imagery. These models can learn richer features from the time series that generalize better through time without requiring extensive feature engineering and re-engineering. When coupled with the appropriately structured and filtered data, these models can reveal subtle shifts related to SLR that cannot be uncovered on shorter time spans. This will better equip managers to identify areas of concern and inform coastal resiliency efforts. In the face of climate change, this capability moves from compelling to critical. In our study area in the coastal plain of North Carolina, USA, we showed that the area classified as wetland more than doubled from 1989 to 2011. This encroachment of wetland into areas that were previously farmland and forested areas is a striking example of SLR and emphasizes the need to develop models capable of appropriately monitoring climate impacts on coastal ecosystems.

**Supplementary Materials:** The following are available online at https://www.mdpi.com/article/10.3390/rs13193953/s1, Figure S1: Distance between patches during generation of testing, validation, and training data. Figure S2: Distribution of the elevations of all pixels, Table S1: Architecture for recurrent neural network.

**Author Contributions:** The study was conceptualized by P.C.G., D.W.J., J.T.R. Funding was procured by P.C.G., D.W.J. and J.T.R. Coding and model development were performed by P.C.G. and D.F.C. Data validation for model development was performed by P.C.G. and D.F.C. All authors (P.C.G., D.W.J., J.T.R., D.F.C., H.R.K., E.A.U.) contributed to the interpretation of the results and writing of the manuscript. All authors have read and agreed to the published version of the manuscript.

**Funding:** Funding was provided by the NASA Terrestrial Ecology Program through the Future Investigators in Earth and Space Science Technology, the North Carolina Space Grant Graduate Research Fellowship, and the Duke Bass Connections program. Microsoft AI for Earth supported with compute time on an Azure-based virtual machine where all development was conducted. These sponsors had no role in the conduct of this research or the preparation of the manuscript.

**Data Availability Statement:** Raw and processed data as well as output land cover classifications are be available at https://research.repository.duke.edu/concern/datasets/. All code needed to train, validate, and utilize the model is at https://github.com/patrickcgray/landcover_rcnn. A pre-trained model and a Jupyter notebook for directly using our final trained model are available in the Github repository. The Github repository also includes tools for reproducing this analysis, analyzing land cover change, and processing and properly structuring all data.

**Acknowledgments:** We would also like to thank the full Bass Connections team including Kendall Jeffreys, Sofia Nieto, Yousuf Rehman, and Feroze Mohideen.

**Conflicts of Interest:** The authors declare no conflict of interest.

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
