# Peer review of "Temporally Generalizable Land Cover Classification: A Recurrent Convolutional Neural Network Unveils Major Coastal Change through Time"

_remotesensing, doi:10.3390/rs13193953_

Round 1

Reviewer 1 Report

The paper entitled " Temporally Generalizable Land Cover Classification: A Recurrent Convolutional Neural Network Unveils Major Coastal Change through Time" is interesting and original, however some improvements are needed.

The authors should cite and compare RCNN used against other architectures or at least point out to the reader their existence, e.g.: ,

- a new architecture, called FCdDNet, which is only four months old, and which performs very well in segmentation:

[1] Ouahabi, A.; Taleb-Ahmed, A. Deep learning for real-time semantic segmentation: Application in ultrasound imaging. Pattern Recognit. Lett. 2021144, 27–34. 

- The reference [2] must absolutely be cited because it opens up new avenues for the problem of change detection based on remote sensing data:

[2] Shi, W.; Zhang, M.; Zhang, R.; Chen, S.; Zhan, Z. Change Detection Based on Artificial Intelligence: State-of-the-Art and Challenges. Remote Sens. 202012, 1688. https://doi.org/10.3390/rs12101688

-Ablation analysis is required to validate the adopted architecture.

- Finally, the conclusion is confusing, and the reader loses the very purpose of the work. Instead of focusing on the Landsat program, which has been known for a long time, please clearly state the conclusion, redefine the objective of the work and quantify the contribution of the proposed method and its interest.

Reviewer 2 Report

The authors proposed the use of a recurrent convolutional neural network (RCNN) for the study of coastal change through time. They focused on the coastal plain of North Carolina in the USA where they analysed Landsat-5 image time series over three different periods (1989, 2000, 2011). They compared a set of machine (RF, SVM) and deep learning (2D-CNN, RNN, RCNN) algorithms for the classification into 6 classes of the study area. Results highlight major changes (mainly forests to farms and wetland, and farms to forests).

Strengths

The context and the motivations are well presented from application and technical perspectives. The manuscript is well structured, written, and presented. The discussion also strengthens the manuscript by delivering more objective insights into the work.

Weaknesses

While the motivations are clearly explained, the literature review on the classification of sequences of satellite images misses the most recent advances in the field.  Moreover, the experimental settings are disputable. See below for the detailed comments.

I recommend major revisions.

Literature review

Deep learning has been studied for the last past five years for the classification of satellite image time series. Contrary to what it is presented, CNNs have been successfully used for time series classification, by turning 2D convolution to 1D temporal convolution, with state-of-the-art results for general-purpose classification tasks [A], and high accuracy in remote sensing (see for example [B], which uses 1D-CNN for crop type mapping).  Moreover, RCNN (also known as ConvLSTM) has already demonstrated its high accuracy for land cover classification [C]. Please note also there exist other ways to learn spatio-temporal features from satellite image time series (ConvGRU+CNN [D], GRU+CNN [E], or U-Net-like network [F]). Meanwhile, RNN and its variants have been extensively applied to optical and radar satellite image time series [G-J]. Finally, CNN and RNN are presented as the two main architectures, hiding the last advances around attention mechanisms and the well-known Transformers, which already have been successfully tested and adapted to satellite image time series [K-M].

Experiments

The training, validation, testing split procedure described in section 2.2.2 is not appropriate for land cover classification, especially for object-based classification. First, the patches extracted need to have no overlap between the three sets, which can only be ensured by using pixels apart of at least 18 pixels in the current setting (patches with a size of 9 pix x 9 pix). Second, even with this strategy, labelled pixels belonging to different sets need to be extracted from different objects (polygons) in the reference data (here NLCD), otherwise, the classification is artificially high. When using image patches at the inputs of a deep learning approach, the best splitting strategy is probably to use a grid as proposed in [C, Figure 4].

In addition, there is currently no discussion about the design of the different architectures (number of hidden layers or units, optimisation strategy, loss used, etc.) While I recognize that hyperparameter tuning is time consuming for sometimes a low outcome (section 4.2), it is still important to assure equity when comparing the different approaches. One solution would be to have a similar number of trainable parameters for the different deep learning models.

Finally, the rationale behind RCNN-LSTM is not explained, and I am not convinced by this type of architecture. What is the advantage to add an LSTM layer compared to have an additional ConvLSTM layer?

Recommendation on the experimental settings

  1. While evaluating RF and SVM on the same data as RCNN is valuable, I also recommend evaluating their performance by only using the spectro-temporal information of each labelled pixel. The goal is here two folds: (i) observing if reducing the feature space dimension is not more favourable to these traditional approaches, that are sensitive to Hughes’s phenomenon, and (ii) concluding on the benefit of the spatial information.
  2. Following the discussion on the literature review, I suggest comparing also the results to a temporal CNN architecture

Other comments:

  • The result section (3.1) indicates that “model accuracy in 2011 was not a good indicator of temporal generalizability”. Please elaborate on this statement. Why it is not a good indicator, and what are your conclusions regarding the results displayed in Figure 5 where the 2D-CNN model outperforms RCNN.
  • Improve the quality of all the Figures as they are blurry. This is a major issue, especially when displaying the results. Figure 5 cannot be interpreted correctly and should be turned into a table of the results.
  • Section 2.2.1. “The same normalization values and calculation were applied to 2000 and 1989”. How did you take into account the different acquisition dates between the three years?
  • Figure 2. Please display all figures in false color (as it was done in Figure 3).
  • Figure 4: colors are not correctly associated between both panels
  • Section 2.2.1 “and 2011 was the last year meeting these requirements”. You can maybe add that Landsat-5 was decommissioned in 2013 (and NLCD was produced in 2011.
  • Section 2.2.1 “using bands 1, 2, 3, 4, 5, and 7”. 6 is missing
  • What is the “developed” class?
  • Sections 1.2 and 2.1. There exist some issues with the citation format.
  • References [14] and [46] are identical.

References

[A] Fawaz, H. I., Forestier, G., Weber, J., Idoumghar, L., & Muller, P. A. (2019). Deep learning for time series classification: a review. Data mining and knowledge discovery, 33(4), 917-963.

[B] Zhong, L., Hu, L., & Zhou, H. (2019). Deep learning based multi-temporal crop classification. Remote sensing of environment, 221, 430-443.

[C] Rußwurm, M., & Körner, M. (2018). Multi-temporal land cover classification with sequential recurrent encoders. ISPRS International Journal of Geo-Information, 7(4), 129.

[D] Ienco, D., Interdonato, R., Gaetano, R., & Minh, D. H. T. (2019). Combining Sentinel-1 and Sentinel-2 Satellite Image Time Series for land cover mapping via a multi-source deep learning architecture. ISPRS Journal of Photogrammetry and Remote Sensing, 158, 11-22.

[E] Interdonato, R., Ienco, D., Gaetano, R., & Ose, K. (2019). DuPLO: A DUal view Point deep Learning architecture for time series classificatiOn. ISPRS journal of photogrammetry and remote sensing, 149, 91-104.

[F] M Rustowicz, R., Cheong, R., Wang, L., Ermon, S., Burke, M., & Lobell, D. (2019). Semantic segmentation of crop type in Africa: A novel dataset and analysis of deep learning methods. In Proceedings of the IEEE/CVF Conference on Computer Vision and Pattern Recognition Workshops (pp. 75-82).

[G] Minh, D. H. T., Ienco, D., Gaetano, R., Lalande, N., Ndikumana, E., Osman, F., & Maurel, P. (2018). Deep recurrent neural networks for winter vegetation quality mapping via multitemporal SAR Sentinel-1. IEEE Geoscience and Remote Sensing Letters, 15(3), 464-468.

[H] Rußwurm, M., & Korner, M. (2017). Temporal vegetation modelling using long short-term memory networks for crop identification from medium-resolution multi-spectral satellite images. In Proceedings of the IEEE Conference on Computer Vision and Pattern Recognition Workshops (pp. 11-19).

[I] Rußwurm, M., & Körner, M. (2017). Multi-Temporal Land Cover Classification with Long Short-Term Memory Neural Networks. ISPRS-International Archives of the Photogrammetry, Remote Sensing and Spatial Information Sciences, 42, 551-558.

[J] Ndikumana, E., Ho Tong Minh, D., Baghdadi, N., Courault, D., & Hossard, L. (2018). Deep recurrent neural network for agricultural classification using multitemporal SAR Sentinel-1 for Camargue, France. Remote Sensing, 10(8), 1217.

[K] Rußwurm, M., & Körner, M. (2020). Self-attention for raw optical satellite time series classification. ISPRS Journal of Photogrammetry and Remote Sensing, 169, 421-435.

[L] Garnot, V. S. F., Landrieu, L., Giordano, S., & Chehata, N. (2020). Satellite image time series classification with pixel-set encoders and temporal self-attention. In Proceedings of the IEEE/CVF Conference on Computer Vision and Pattern Recognition (pp. 12325-12334).

[M] Yuan, Y., & Lin, L. (2020). Self-Supervised Pretraining of Transformers for Satellite Image Time Series Classification. IEEE Journal of Selected Topics in Applied Earth Observations and Remote Sensing, 14, 474-487.

Round 2

Reviewer 1 Report

The authors used a well-known architecture based on Recurrent Convolutional Neural Network and did not introduce anything new in this respect. It is imperative that they consider an opening to enrich their manuscript by citing new and very promising works that propose an encoder-decoder architecture such as FCdDNet [1] and Transformer [2]

[1] Ouahabi, A.; Taleb-Ahmed, A. Deep learning for real-time semantic segmentation: Application in ultrasound imaging. Pattern Recognit. Lett. 2021144, 27–34

[2] Salman Khan, Muzammal Naseer, Munawar Hayat, Syed Waqas Zamir, Fahad Shahbaz Khan and Mubarak Shah. Transformers in Vision: A Survey. 2021, 2101.01169. arXiv cs.CV

Author Response

Thank you for this quick response and both helpful citations. These two references have been properly integrated into section 1.2 and add a helpful view of both FCdDNets and Transformers, future tools which may improve landcover predictions across time.